



# Impact of high and low vorticity turbulence on cloud environment mixing and cloud microphysics processes

Bipin Kumar[1], Rahul Ranjan[1,2], Man-Kong Yau[3], Sudarsan Bera[1], and Suryachandra A. Rao[1]

[1]Indian Institute of Tropical Meteorology, Ministry of Earth Scinces, Homi Bhabha Road, Pashan, 411008, India
[2]Department of Atmospheric and Space Science, Savitribai Phule Pune University, Pune, India 411007, India
[3]Department of Atmospheric and Ocean Science, McGill University, Burnside 805 Sherbrooke Street Montreal, Quebec, Canada H3A 0B9

**Correspondence:** Bipin Kumar (bipink@tropmet.res.in)

**Abstract.** Turbulent mixing of dry air affects evolution of cloud droplet size spectrum through various mechanisms. In a turbulent cloud, high and low vorticity regions coexist and inertial clustering of cloud droplets can occur in a low vorticity region. The non-uniformity in spatial distribution of size and number of droplets, variable vertical velocity in vortical turbulent structures, and dilution by entrainment/mixing may result in spatial supersaturation variability, which affects the evolution
of the cloud droplet size spectrum by condensation and evaporation processes. To untangle the processes involved in mixing phenomena, a direct numerical simulation (DNS) of turbulent-mixing followed by droplet evaporation/condensation in a submeter-cubed-sized domain with a large number of droplets is performed in this study. Analysis focused on the thermodynamic and microphysical characteristics of the droplets and flow in high and low vorticity regions. The impact of vorticity production in turbulent flows on mixing and cloud microphysics is illustrated.

**Keywords:** Droplet characteristics, High and low vorticity regions, Cloud turbulence, k-means clustering, Degree of mixing.

**Highlights:**

- Regions of high vorticity are prone towards homogeneous mixing due to faster mixing by air circulation.
- Microphysical droplet size distribution is wider (narrower) in high (low) vorticity volumes.
- Drier environmental air mixing leads initially to higher spectral width and decay later on.
- k-means clustering algorithm (an unsupervised machine learning technique) is used to identify regions of high vorticity in the computational domain.

## 1 Introduction

Clouds are visible accumulations of tiny water droplets or ice crystals in the Earth's atmosphere. They exhibit multiple roles in
atmospheric processes, ranging from the radiation budget to the hydrological cycle (Bengtsson, 2010; Grabowski and Petch, 2009; Harrison et al., 1990; Randall and Tjemkes, 1991). The size of clouds may extend from a few meters to several kilometers. However, the suspended droplets that constitute a cloud are much smaller and typically of size 1-20 $\mu m$ in radius. The journey from a cloud droplet to a raindrop ($\sim 10^3 \mu m$) is a complicated process. Condensation, collision and coalescence, the three





key processes involved in the growth of a droplet, are prominent at different stages of a cloud (Rogers and Yau, 1996). For

example, up to 15 $\mu$m, diffusional-condensation growth dominates, while collision-coalescence is effective when the droplet

radius reaches approximately 40 $\mu$m (Pruppacher, 2020). The rapid growth of droplets in size range 15-40 $\mu$m in radius for

which neither condensational growth nor the collision-coalescence is effective, is poorly understood. This size range has been

termed as the condensation-coalescence bottleneck or the size gap (Grabowski and Wang, 2013). The rapid growth of droplets

in the 'size gap' is regarded as one of the important unresolved problems in cloud physics. To explain the rapid growth, several

mechanisms like entrainment, turbulent supersaturation fluctuations, enhanced collision rates due to turbulence and the role of

giant aerosols have been proposed. This study focuses on the interaction between droplets and turbulence to explore how this

interaction modifies the droplet characteristics.

## 1.1 Role of Turbulence in Cloud Microphysics

Understanding the impact of turbulence on the dynamics and microphysics of clouds is a long-standing problem (Devenish et

al., 2012; Khain et al., 2007; Shaw, 2003; Vaillancourt and Yau, 2000) and it has been an active area of research with in situ

observation and numerical models being applied to investigate this multiscale problem. Movement and position of droplets

are controlled by turbulent eddies of varying sizes. Simultaneously, evaporation or condensation of droplets incurs changes in

the local environment ($\sim$ scale of droplet itself) through latent heat exchange. Buoyancy generated by phase change (when a

bunch of droplets evaporate or condense instead of a few) may impact cloud-scale motions. A quantity called particle response

time '$\tau_p$' determines how quickly droplets respond to the changes in the surrounding fluid motion. Some droplets are tiny and

precisely follow the flow trajectory, while larger ones can modify the flow. Thus, droplet-turbulence interaction is a multi-scale

process and non-local in nature.

There are several macrophysical and microphysical implications of droplet and turbulence interactions. Droplets in the

decaying part of a cloud may be transported by turbulence to the more active regions of the cloud and undergo further growth

(Jonas, 1991). The inhomogeneous mixing model used by Cooper et al. (1986) and Cooper (1989) shows that when a parcel

of air undergoes successive entrainment events, each of which reduces the droplet concentration, enhanced growth is possible.

However, Jonas (1996) argues that ascent leading to droplet growth may activate some entrained nuclei, limiting the maximum

supersaturation achieved which, in turn limits the growth rate. Vaillancourt et al. (1997) gives further insight into the nature

of turbulent entrainment at the cloud edges. They argue that the interaction between the ambient and the cloudy air is not the

same everywhere rather there is some prominent regions of entrainment with vortex circulations.

The study of microphysical droplet turbulence-interaction has gained momentum in the recent years due to advances in

computational capabilities. Several possibilities like turbulence induced supersaturation fluctuation and enhanced collision rates

have been investigated. Some studies (Chen et al., 2016; Franklin et al., 2005; Pinsky et al., 2000; Riemer and Wexler, 2005;

Shaw, 2003; Vaillancourt and Yau, 2000) indicate an enhanced collision rate in a turbulent environment. Shaw et al. (1998)

performed DNS and found that preferential clustering of droplets in the low vorticity regions in a cloud gives rise to spatially

varying supersaturation. Droplets in the high vorticity regions experience enhanced supersaturation and grow faster. However,

the comments of Grabowski and Vaillancourt (1999) on the results of Shaw et al. (1998) pointed several shortcomings, in



particular, absence of droplet sedimentation, assumption of high volume fraction of vortex tubes (50%) and strong dependence on the vortex lifetime.

There is no clear theory regarding vorticity characteristics in three-dimensional homogeneous turbulent flows, despite increasing research on turbulence. Vorticity has a profound impact on the spatial distribution of droplets. Due to preferential clustering, a relatively less number of droplets are left in the high vorticity regions. The difference in the spatial distribution of droplets induces supersaturation fluctuations. A low number of droplets competing for the available vapour field in the high vorticity regions, should experience enhanced growth rates and for this to happen, droplets should stay there for duration

enough for supersaturation field to act. However, little is known about the length and lifetime that the high vorticity regions occupy (Grabowski and Wang, 2013). Due to these limitations, the effect of preferential clustering on the diffusional growth is poorly understood.

In this study, we examine the diffusional growth and evaporation of cloud droplets in an entrainment and mixing simulation setup of DNS. We compared the droplet characteristics such as spectral width, volume mean radius, number concentration,

probability density function of droplet radii and supersaturation in high and low vorticity regions. As reported by Vaillancourt et al. (1997), the main entrainment sites and mixing zones were located in the vortex circulations areas. Similar to Vaillancourt et al. (1997), we aims to look for, using DNS, locations with vortex circulations in the main entrainment sites and mixing zones. The organisation of the paper is as follows. The next section provides details of methods employed and data used. Results and discussions are provided in section 3 with further four subsections containing discussion of flow and droplet characteristics in

low and high vorticity regions. In the last part, we concluded our analysis.

## 2    Data and Methods

We carried out a Direct Numerical Simulation (DNS) following the setup of Kumar et al. (2014, 2018) to simulate the entrainment and mixing mechanisms at cloud edges. This DNS code uses the Euler-Lagrangian frame, solves flow equations at each grid point, and track each droplet inside a grid by integrating equations for their position, velocities, and growth rate. The

simulation produces output in two formats, one is from the Eulerian frame in NetCDF format developed by UCAR/Unidata and the droplet dynamics output is saved in SION format (SIONLib, 2020). The simulation domain was chosen $(51.2\,cm)^3$ with $1\,mm$ grid resolution, thus containing a total $(512)^3$ grid points in the domain. An initial setup of computational domain is presented by the Figure 1(a).

Four simulation setups were considered for this study. Two relative humidity (RH) set up (85% and 22%) cases with both

mono-dispersed and poly-dispersed droplet size distribution initialized to DNS. The mono-dispersed case uses a single droplet size of 20 $\mu$m (an idealistic case), whereas the poly-dispersed set up uses droplet size distribution (size range 2-18 $\mu$m) from cloud observation (CAIPEEX experiment: https://www.tropmet.res.in/ caipeex/ ) similar to earlier simulations as used in Kumar et al. (2017). Two humidity cases correspond to dry (RH=22%) and moist environmental (RH=85%) airs were taken (from observations of the monsoon environment) with a cloudy slab in the DNS domain to simulate the mixing processes.





This study aims to investigate droplet characteristics in high and low vorticity regions of cloud turbulence. Therefore, the vorticity magnitudes were first calculated using Eulerian data at each grid point containing the velocity components in X, Y & Z directions. The next step is to find the high and low vorticity region in the DNS computational domain, requiring calculating and visualizing actual vortices generated by turbulence flow. Since the grid size is $1\,mm$, it is unfeasible to get a vortex inside a single grid box; instead, an area of the vortex must be sought out, containing multiple grid boxes. It is challenging task to locate

a small box to cover a minimal portion of the low vorticity area. We used an unsupervised machine learning (ML) algorithm mentioned in the next subsection to address this problem.

### 2.1    Locating High Vorticity Regions

To locate high vorticity regions in the domain, we used the k-means clustering algorithm from Scikit-Learn python package (Pedregosa, 2011). The k-means clustering (Bock, 2007) is one of the most popular and the simplest unsupervised machine

learning algorithm. It makes 'k' groups or clusters from a dataset based on the Euclidian distance between individual data points. However, k-means clustering cannot guess the optimum number of clusters for a particular dataset; instead, the user has to assign it. Therefore, selecting the number of groups or clusters in which a dataset has to be grouped or clustered is crucial. This algorithm was used to locate high vorticity region from the vorticity data.

The absolute vorticity $\omega$ related to velocity component is calculated as (see Chapter 4.2 in Holton and Hakim (2013))

$$\omega = (\omega_i^2 + \omega_j^2 + \omega_k^2)^{\frac{1}{2}} \tag{1}$$

where

$$\omega_i = \frac{\delta w}{\delta y} - \frac{\delta v}{\delta z}; \qquad \omega_j = \frac{\delta u}{\delta z} - \frac{\delta w}{\delta x}; \qquad \omega_j = \frac{\delta v}{\delta x} - \frac{\delta u}{\delta y} \tag{2}$$

The values of $\omega$ in our DNS data range from $0 - 200\,s^{-1}$ as seen in panel (a) of Figure 1. Vaillancourt and Yau (2000) documented that only small fraction is occupied by high vorticity regions and no preferential concentration of cloud droplet

was observed in cloud core. We also found that less than 2% of grids (by volume) having vorticity magnitude $60 s^{-1}$ while for bigger magnitude even less number (almost negligible ) of grids were located. Based on these findings, in this work, a threshold value of vorticity magnitude, $60 s^{-1}$ was chosen as high vorticity criteria. We have investigated the droplets characteristics taking $50 s^{-1}$ as threshold but it did not make any difference in the trends. Considering $30 s^{-1}$ as a threshold for high vorticity, which is less than $1/5^{th}$ of the maximum vorticity magnitude ($200 s^{-1}$) does not seem to be justified. Figure 1(b) depicts the

fraction of grid points occupied for different threshold values of vorticity magnitude.

Once the threshold value for vorticity magnitude is decided, the next step is to locate 3D boxes enclosing the high vorticity regions which is accomplished by k-means clustering. Wherein, two input variables have to be assigned a value; (i) number of cluster ('k') and (ii) maximum number of iterations. Since, vortexes are having tabular or sheet type structure, it is possible that a 3D box may contains many low vortex regions for a typical value of 'k'. Hence, an optimal value of 'k' is required for





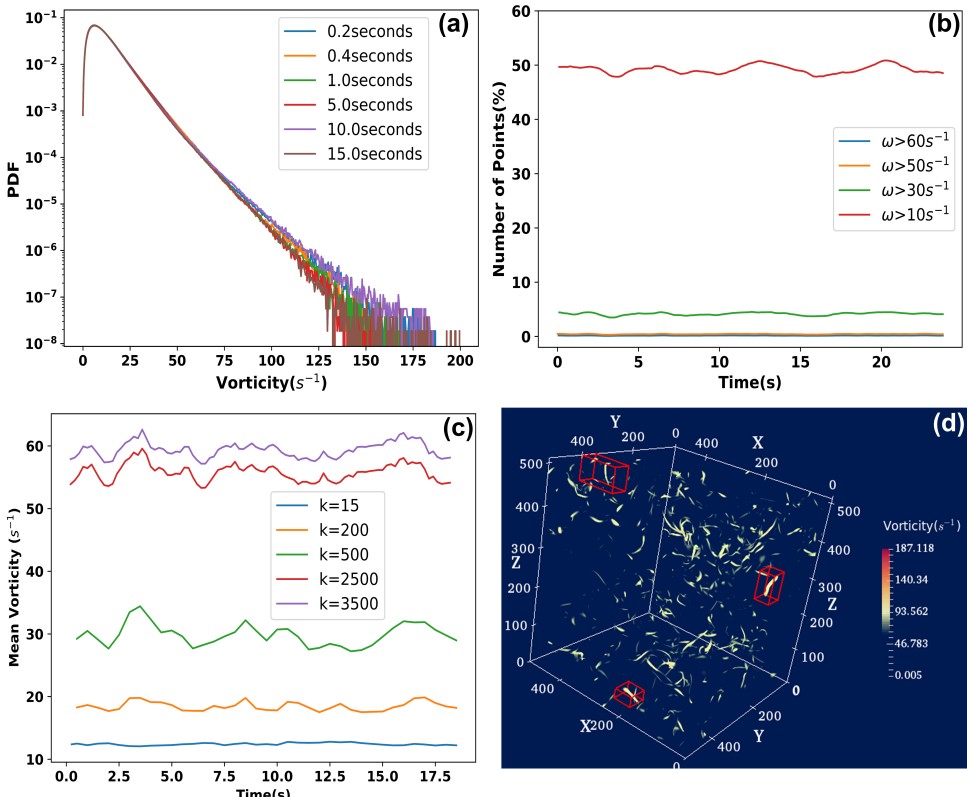

**Figure 1.** The panel (a) displays the PDF of vorticity at different times. The fraction of high vorticity points (based on threshold values) are depicted in panel (b). Panel (c) represents vorticity values for different values of 'k'. Panel (d) shows high vorticity regions (enclosed by cubic boxes) obtained for k=500.

| Set up | Shaw et al. (1998) | This study |
|---|---|---|
| Entrainment-Mixing | No | Yes |
| Vortex Lifetime | 2-3 orders of magnitude > Kolmogorov timescale ($\approx$10 seconds) | Less than 1 second |
| Volume fraction of high vorticity | $\approx 50\%$ | less than 2% |

**Table 1.** Comparison of findings in this study and in Shaw et al. (1998). Kolmogorov time scales in natural clouds are in the range of $0.01 \rightarrow 0.1$ seconds. Kolmogorov time scale for DNS is 0.0674 seconds, which lies in the said range.

choosing small enough 3D boxes to avoid the low magnitude vortexes. We identified the value of 'k' to be used in the algorithm by conducting several experiments and selected optimal value k=3500 based on the chosen threshold value for high vorticity. Figure 1(c) provides vorticity values for different 'k' values which confirms the choice of k=3500 in this study. Similarly, number of iterations were chosen as 200. Any structure representing a region of high vorticity or a vortex was considered a cluster.






A typical visualization of finding a box enclosing a high vorticity region is shown in the Figure 1(d), where a cubical box is shown to surround the high vorticity region. It was noted that high vorticity regions occupy only a tiny fraction (0.1- 0.2 %) of the total domain.

This finding of a tiny fraction for high vorticity regions in a cloud core is significant because it is completely different from
the results documented in Shaw et al. (1998). They hypothesizes that preferential concentration ( inertial clustering) occurs at small spatial scales and low (high) particle concentration corresponds to high (low) vorticity regions. Furthermore, they used Rankine vortex model and did not calculate vorticity from the velocity field of the DNS like we did in this study. A comparison of methods in this work and study done by Shaw et al. (1998) is provided in the table 1.

## 3    Results and discussion

In this section, we discuss about various analyses from the two humidity set up cases with initial poly-dispersed size distribution is considered.

### 3.1    Turbulence characteristics at the edges and core of cloud

The interface between cloud volume and the sub-saturated air is distinguishable during the early evolution of the flow. To see if any features of the flow exhibit distinct properties at the edges, three separate volumes from the entire domain have been
picked up. The cloudy slab area lies between 142mm to 372mm along x-axis, and the rest is occupied by the sub-saturated air. Of the two interface volumes, one is on the left side (between $x = 70\,mm$ to $140\,mm$), and the other is on the right side ($x = 364\,mm$ to $434\,mm$). The volume lies between $x = 182\,mm$ and $322\,mm$ is for the core region, as depicted in Figure 2.

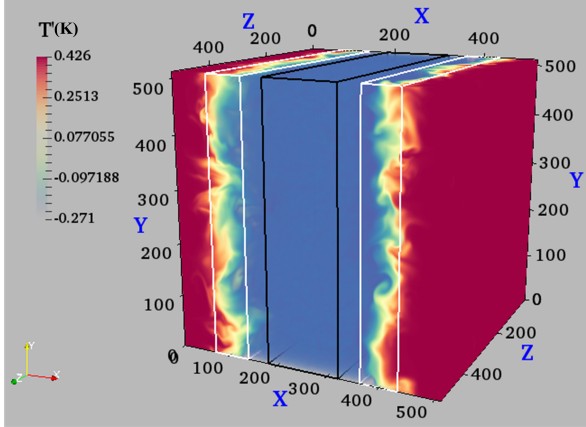

**Figure 2.** The snapshot DNS domain at 0.2 seconds time which can be considered as initial state of the simulation. It also shows two boxes, particularly at the edges (left and right white boxes) and the initial cloud slab (central black box). The legends represent fluctuations from mean temperature.





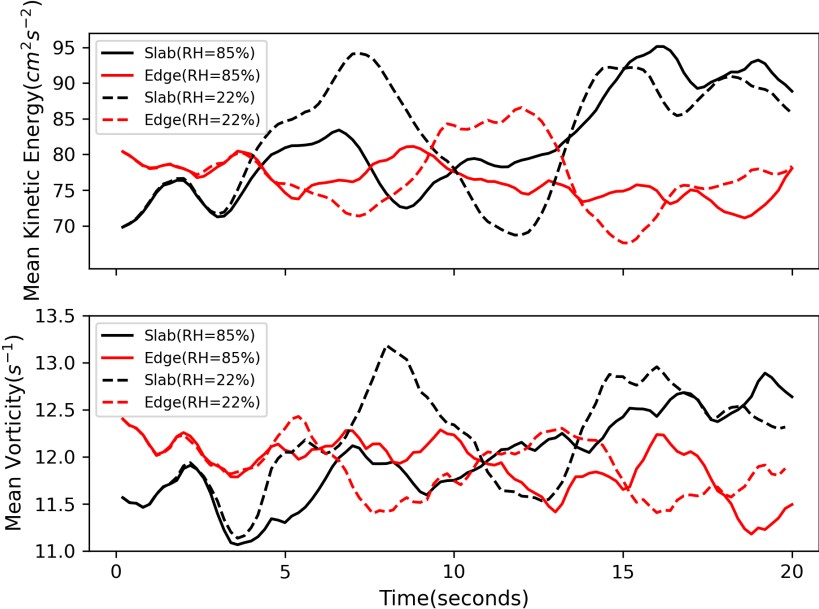

**Figure 3.** Upper panel is for the variation in kinetic energy (KE) for both drier and humid cases in the cloudy slab and edges. Mean vorticity variations at the edges and within the cloudy slab, in both cases, is shown in the lower panel.

The cloudy volume properties initially see sharp changes, which was confirmed by analysing the variations in kinetic energy (KE), vorticity, and mixing ratio. The edges are the most turbulent part of clouds during early evolution time; thus, a more

considerable amount and robust fluctuations in KE are experienced there, depicted in the upper panel of Figure 3. The availability of more kinetic energy at the cloud edges makes them the hotspots of vorticity generation. Initially, a higher value of mean vorticity is observed at the edges in both simulation cases, as evident from the lower panel of Figure 3. Near the edges, a strong gradient of mixing ratio and temperature exists, which leads to a turbulent mixing process. TKE is generated at the interface of cloudy slab and dry air due to negative buoyancy production by droplet evaporation. This energy is transported

to cloud slab with progressive time by the vortices ('eddies') that propagate inside. However, there are periodic changes in the TKE variations between the interface and the cloud slab possibly due to periodic boundary condition of the DNS setup. Notable feature between the two RH simulations (22% and 85%) is higher difference between the interface and the cloud slab as it undergoes stronger droplet evaporation in the drier case (RH=22%).

### 3.2 Flow characteristics in high and low vorticity regions

The previous subsection presented the variation in flow characteristics at cloud core and edges. Another part of this study is to investigate these characteristics in high vortex regions of the turbulent flow. We considered the high vorticity (HV) area having a vorticity magnitude greater than $60\,s^{-1}$. Similarly, points with vorticity of less than $30\,s^{-1}$ were classified as regions of low vorticity (LV).





We investigated the evolution of the mixing ratio (qv) in both drier (RH=22%) and humid cases (RH=85%). The incursion of

drier air results in a lower mixing ratio at the edges. In both HV and LV regions, apart from the mixing ratio, we also determined the root mean square velocity $u_{rms}$. For dry and humid cases, the $u_{rms}$ is found to be higher in HV regions. We examined the droplet features for RH=22% and RH=85% after investigating the flow properties described in the next subsection.

### 3.3 Droplet Characteristics

One of the main aims of this study is to study various droplet characteristics such as number concentration, volume mean

radius, spectral width, and mixing process in HV and LV regions.

#### 3.3.1 Number concentration and mean volume radius

There have been many kinds of research on the distribution of droplets in a turbulent flow field. Several laboratory studies (Lian et al. , 2013) and model simulations (Shaw et al., 1998; Ayala et al., 2008) reported the process of preferential clustering of cloud droplets in low vorticity region. The preferential clustering means that droplets prefer to cluster in some specific flow

regions rather than randomly distributed everywhere. A high amount of rotation characterizes the highly vortical part of a fluid. When a droplet enters this region, it flung out due to its inertia and accumulated in a low vorticity region. This process leads to heterogeneous droplet concentration in space, an important aspect that affects droplet growth rate and size distribution. In a poly-dispersed droplet size distribution, the larger droplets are more prone to be affected by the vorticity compared to smaller droplets that may follow the flow streamline due to low inertia. For this reason larger droplets accumulate in low vorticity

region and results in larger mean volume radius.

On the upper panel of Figure 3, both humidity cases show almost the same trend, i.e., a higher number concentration in the low vorticity region due to inertial clustering. Lower panel of Fig. 3 shows variation of mean volume radius in high and low vorticity regions. Arid like condition in the case with RH=22% leads to quick evaporation of droplets and indicated by rapid decay in droplet number concentration and mean volume radius. Consequently, the number concentration curves almost merge

after 7.5 seconds. Due to preferential clustering, the high vorticity regions have a relatively small number of droplets. It is to be noted that low vorticity region always has larger mean volume radius during the simulations as shown in the lower panel of Figure 4.

There may be two possible reasons for smaller value of mean radius in high vorticity regions: (i) droplets experience a drier environment and getting more evaporation during early evolution of mixing when high vorticity forms at cloud edge, and (ii)

larger droplets flung out of the high vorticity region more easily as a consequence of greater inertia effect, leaving behind only the smaller ones, i.e., preferential clustering is more prominent for larger droplets. The second possibility is more valid during later part of simulation (approx. after 5 sec) when high vorticity forms inside the cloud slab. To clarify whether preferential clustering alone decides the volume mean radius distribution or any other mechanism that may be responsible, we investigated droplet spectra, the trends of mean supersaturations, and evolution of droplet size distribution.



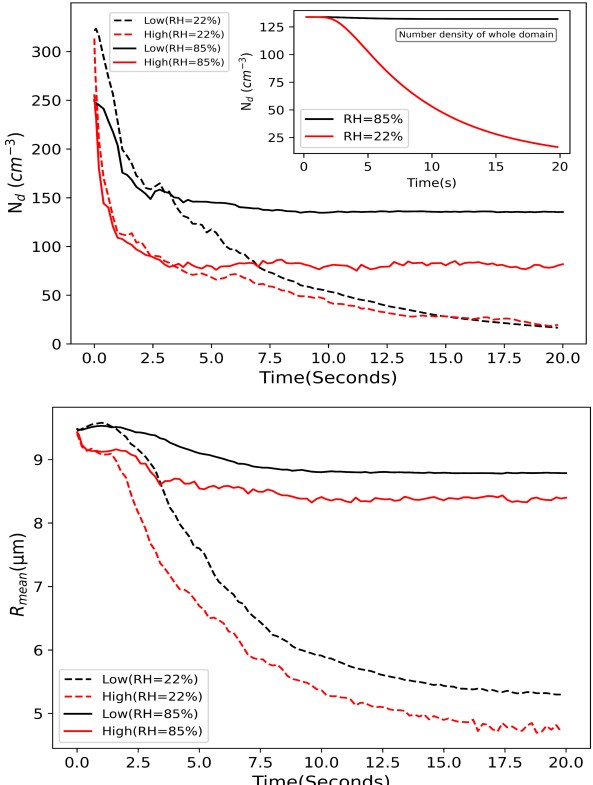

**Figure 4.** Evolution of number concentration ($N_d$) in the whole domain (the inner panel) is shown in the upper panel. In both cases, initial number densities (in the entire domain) were close to $130\,cm^{-3}$. However, the evolution of Nd in high and low vorticity regions have different magnitudes. It is always greater in the low vorticity regions. The lower panel shows the volume mean radius which is always smaller in the high vorticity regions.

### 3.3.2 Evolution of Droplet Size Spectra

The variation of the spectral width is presented in Figure 5, showing an entirely different picture of the evolution of droplet spectra in two cases. We have initialized the DNS with a poly-dispersed DSD spectra having a spectral width of nearly 2.2 $\mu$m, which is observed in monsoon cumulus clouds over India (Bera et al., 2016).

During the mixing of cloud slab with the environmental dry air, the spectral width of DSD increases rapidly initially from 5-7 seconds and decreases thereafter for RH22% case. Whereas in the RH85% case, a gradual increase can be seen for initial 10 seconds and remains almost constant after that. One of the most important results of this study is that spectral width of DSD is different in high and low vorticity regions. For RH22% case, spectral width is higher for droplets situated in high vorticity region during initial 5 seconds of mixing when spectral width increases rapidly. But opposite scenario occurs after 7.5 seconds i.e., smaller spectral width in high vorticity region. Nevertheless, spectral width always remains higher in high vorticity region in RH85% case.





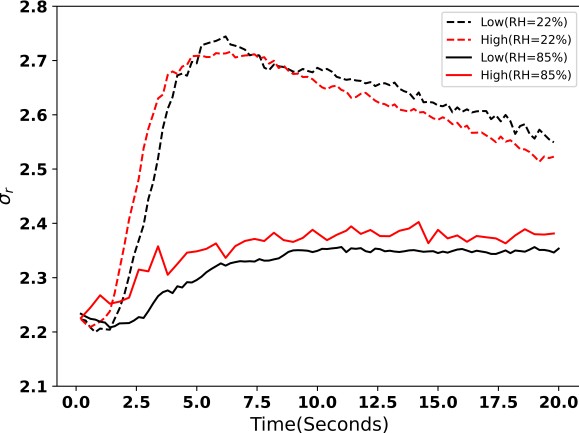

**Figure 5.** Variation of spectral width in high and low vorticity region. The spectral width is always greater in the high vorticity region in the RH85% case while it is greater for the initial 5 seconds only in RH22% case.

The initial growth and later decay of spectral width during mixing for RH22% case is associated with the modification of spectral shape by droplet evaporation and number concentration dilution (see Bera (2021)). In this case, evaporation is very significant due to much drier environmental air mixing. When evaporation starts, smaller droplets of the DSD evaporate faster compared to larger droplets governed by the inverse relation of growth rate with droplet size (Rogers and Yau, 1996). As a

result, spectra broaden towards the smaller size tail as shown in upper panel of Fig. 6. This is the reason of increasing spectral width for initial 5 seconds of RH22% case and entire 20 seconds of RH85% case. However, when evaporation is so sufficient that smaller size tail of spectra is evaporated completely and larger droplets only remain to evaporate, the spectra start shrinking and spectral width decreases (as shown in Fig. 6c). This is the situation for RH22% during mixing after 7.5 seconds and this situation does not occur for RH85% case where evaporation is not sufficient due to moist air mixing (as shown in Fig. 6d).

The difference of spectral width in HV and LV region can be explained with consideration of higher droplet evaporation in high vorticity region. Initially, high vorticity forms at cloud edges where dry air mixing occurs, leading to faster evaporation of droplets. The second possibility is that high vorticity regions are pockets of rotating air motions that can easily transport the vapour mass (produced by droplet evaporation) out of the region and thereby facilitates enhanced evaporation. These two plausible reasons result in higher evaporation rate in regions of high vorticity and consequent impact of droplet spectral width.

The differences in the PDFs of HV and LV regions can be noticed very well at 3 and 17.8 seconds. During this time, a greater spectral width exists in the high vorticity region (refer to figure 4) for both humidity cases (i.e., RH=22% and RH=85%).PDFs confirm that high and low vorticity regions contain almost the same maximum and minimum drop sizes, but the difference comes from the distribution. So, the following possibilities arise based on what PDFs are depicting:

- Supersaturation is lower in high vorticity regions, which triggers enhanced evaporation.
- Due to enhanced evaporation, there are smaller droplets in the vortices as depicted by the size distribution.
- Bigger droplets are more vulnerable to be thrown out of the vortices, leaving only the smaller ones.



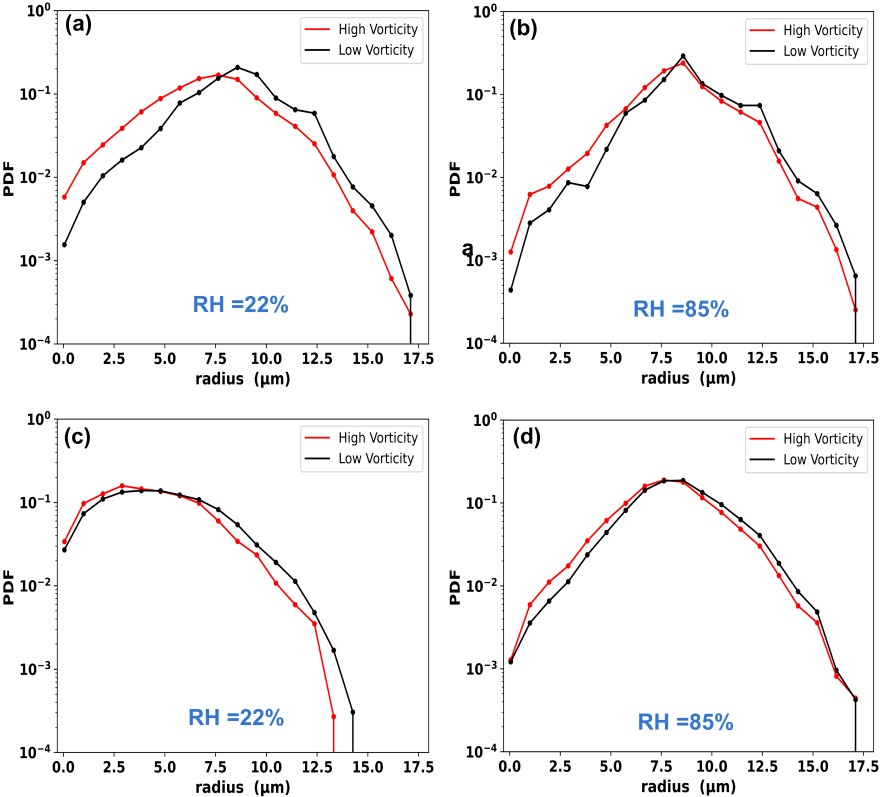

**Figure 6.** Probability density function of droplet radii for 22% RH (left panel) and 85%RH (right panel). Upper panel depicts the plots for 3 seconds and the bottom one is for 17.8 seconds.

For better understanding, we also analysed the trend of the droplet supersaturation.

### 3.3.3 Mean supersaturation

Figure 7 depicts the mean and standard deviation of supersaturation variation in HV and LV vorticity regions for RH=22%
and 85% cases. The droplets in high vorticity regions experience comparatively lower supersaturation until around 6 seconds,
after which the difference tends to vanish. Hence, during entrainment of drier air into the cloudy volume, droplets encounter a
more sub-saturated environment in the highly vortical regions, and it is the lower supersaturation values that produce a larger
standard deviation. In RH22% case, the supersaturation drops to -15%, while in RH85% case, although a similar pattern exists,
the supersaturation drops only up to -3%.

### 3.3.4 Degree of mixing

One of the best metrics to investigate the entrainment and mixing process is the degree of mixing, which depends on the mixing
diagram and has wider application in numerical models (Lehmann et al., 2009; Kumar et al., 2017, 2018). We have analysed

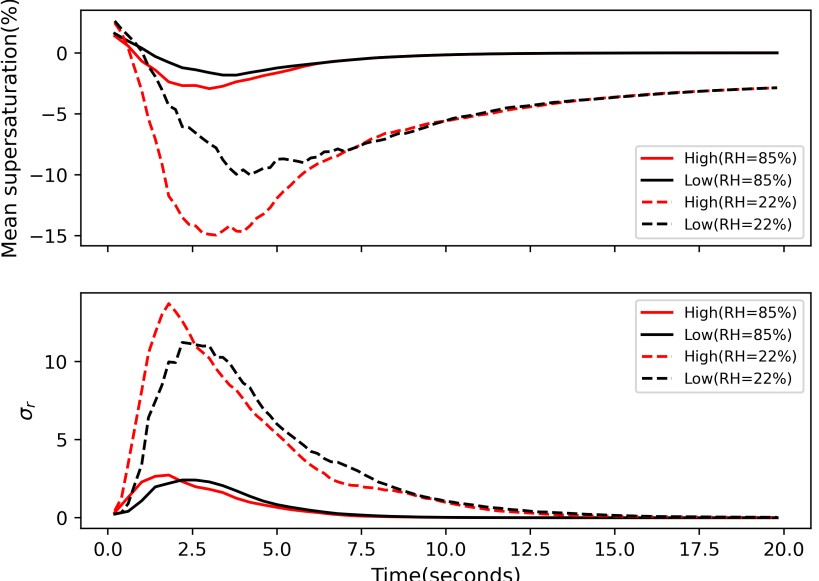

**Figure 7.** Upper panel is for the evolution of mean supersaturation in high and low vorticity regions. Supersaturation is lower in the high vorticity regions during initial 5-7 seconds and its value reaches a minimum of -15% in the RH22% case, while it drops only upto -3% in the RH85% case. Evolution of the standard deviation of supersaturation in high and low vorticity regions is shown in lower panel. The standard deviation sees a steeper increase and decrease in high vorticity regions, and has a large variation (0-14) in RH22% case, while the variation is small (0-2.8) in the RH85% case.

the mixing diagrams and degree of mixing in the high and low vorticity regions for both moist and dry cases. Variations in mixing diagrams in high and low vorticity regions for RH22% case are depicted in the panels (a) and (b) respectively of Figure

7. The panel (c) shows the evolution of the degree of homogeneous mixing and a comparison of Damkohler numbers for all four cases is presented in the panel (d).

The Damkohler number also measures the degree of mixing as a quantity related to two time scales, namely, fluid time scale ($\tau_{fluid} = L/U_{rms}$) and phase relaxation time scale ($\tau_{phase} = \frac{1}{4\pi n_d Dr}$) (Kumar et al., 2012), where, $U_{rms}$ is root-mean-square of the turbulent velocity fluctuation, L is a characteristic large (energy injection) scale of the flow, $n_d$ is droplet number density,

'D' is modified diffusivity, and 'r' denotes initial volume mean radius.

The Damkohler number, $Da = \tau_{fluid}/\tau_{phase}$, represents an estimation of the mixing scenario. $Da >> 1$ indicates an inhomogeneous process, while $Da << 1$ represents a homogeneous one (Latham and Reed, 1997). In the panel (d) of Figure 8, the evolution of the Damkohler number has been shown. Low vorticity regions always have a bigger Da than high vorticity one. A value closer to 0 indicates a higher degree of homogeneous mixing. Like the mixing diagrams, the Damkohler number also

suggests a greater homogeneous mixing in the high vorticity regions. High vorticity (i.e., circulations of fluid) indeed helps to promote faster mixing and produce a well mixed homogeneous cloud volume.





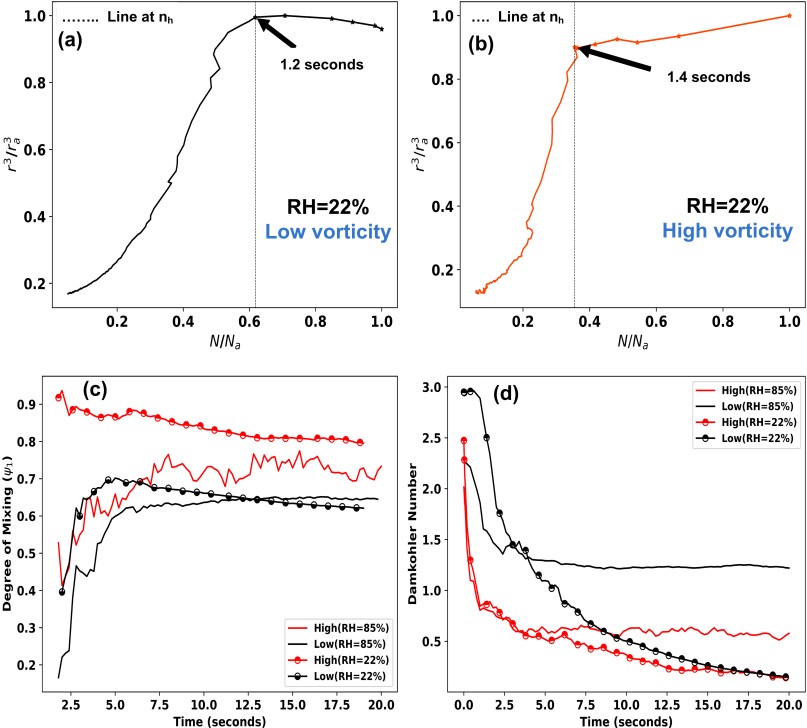

**Figure 8.** The panel (a) and (b) represent the mixing diagrams for low and high vorticity respectively for RH=22% case. Degrees of homogeneous mixing for all four cases are compared in panel (c). A comparison of Damkohler number for all the cases is shown in (d).

## 4 Mono-disperse case

In the previous sections, we have discussed the simulations considering a poly-dispersed spectrum of cloud droplet size distribution based on observation. Here, we have conducted additional simulations with a mono-dispersed droplet size (20 micron) to investigate the effect of vorticity on same sized droplets. The simulation data analysis obtained from the mono-dispersed case yielded quite similar results to the poly-dispersed case. The findings have been summarized in table 2. Due to droplet evaporation associated with entrainment-mixing and condensation growth in slab region, the mono-dispersed spectrum widens and generates various droplet sizes. Higher droplet number concentration and greater mean volume radius is found in low vorticity region, indicative of inertial clustering of larger droplets. Various other properties are also found similar to that of poly-dispersed droplet simulation case as can be noted in table 2.

## 5 Conclusions

Droplet characteristics in high vorticity (HV) and low vorticity (LV) region in a three dimensional DNS of cumulus cloud have been studied. We have taken two initial drop size distributions, one from the CAIPEEX observation (poly-dispersed) (Bera et al., 2016) and the other one is a mono-dispersed size distribution, and performed entrainment simulation with two initial relative humidity values viz. 22% and 85% for the ambient air that mixes with cloud slab.





| Characteristics | Observations in the case with RH=22% | Observations in the case with RH=85% |
|---|---|---|
| **Mixing ratio** | Lower mixing ratio in the high vorticity regions during initial 10 seconds. | Lower mixing ratio in the high vorticity regions during initial 10 seconds. |
| **RMS velocity** | Higher root mean square velocity in the high vorticity regions. | Higher root mean square velocity in the high vorticity regions. |
| **Number Concentration** | The number concentration is always higher in the low vorticity regions. | The number concentration is always higher in the low vorticity regions. |
| **Volume mean radius** | The volume mean radius is always greater in the low vorticity regions. | The volume mean radius is always greater in the low vorticity regions. |
| **Spectral width** | A greater spectral width in high vorticity during initial 5-7 seconds and an unclear trend after that due to the droplets' fast evaporation. | The spectral width is always greater in the high vorticity regions. |
| **Mean supersaturation** | Lower supersaturation in high vorticity regions during initial 5-7 seconds. Supersaturation drops to a minimum value below -15%. | Lower supersaturation in high vorticity regions during initial 5-7 seconds. Supersaturation drops only upto -3%. |
| **The standard deviation of droplet supersaturation** | The greater standard deviation of droplet supersaturation in the high vorticity regions during the initial 2-3 seconds and a reverse trend from 3-7 seconds. No difference thereafter. | The greater standard deviation of droplet supersaturation in the high vorticity regions during the initial 2-3 seconds and a reverse trend from 3-7 seconds. No difference thereafter. |
| **Mixing** | Mixing scenarios estimated through both methods, e.g., mixing diagram and Damkohler number, indicate a more homogeneous mixing in the high vorticity regions. | A higher degree of homogeneous mixing in this case also. |

**Table 2.** Various flow and droplet characteristics of the mono-dispersed case.

A DNS model setup similar to Kumar et al. (2014) has been considered. This setup has a cloudy volume and surrounding sub-saturated air, which are allowed to mix as the entrainment simulation kicks in. During the entrainment and mixing process, the flow in the domain develops spatially varying characteristics. The magnitude of turbulence (decaying with time) is not the same everywhere. Some regions are highly turbulent and possess a high value of vorticity. The vortices may influence the distribution and growth of cloud droplets. To study the dependency of droplet characteristics on vorticity, we located HV regions in the computational domain. Finding HV regions is challenging because the shape, size, and position of vortices change within a fraction of a second. We have applied an unsupervised machine learning algorithm, k-means clustering, to categorize the high and low vorticity clusters. In our knowledge, it is the first time to use the machine learning algorithm for investigating cloud turbulence properties. We answered the following scientific questions in this study

(i) How much volume fraction of intense vorticity occupies in the domain?

(ii) Where is the cloud-clear air interaction most prominent; in highly turbulent regions or weakly turbulent regions?





(iii) Is preferential clustering same for all size of droplets? How do the spectral properties of droplets vary in high and low vorticity regions?

(iv) Does the relative humidity of the ambient air have any impact on the evolution of droplet size spectra?

(v) What is the maximum characteristics, i.e., homogeniety of mixing degree in high vorticity and low vorticity regions?

Entrainment and mixing is a turbulent process, and during the initial few seconds, the cloud edges, where a large gradient of water vapor field exists, are the most turbulent. More robust KE fluctuations were found at cloud edges, making them hotspots for vorticity generation. A distinct difference in the KE fluctuations was noted between two RH simulations (22% and 85%), a
bigger difference was observed in the drier (RH=22%) case. Turbulent velocity urms was found higher in HV regions for both simulation cases.

Droplets tend to cluster in the LV region with smaller droplets showing less tendency for the same, which may lead to heterogeneous number concentration in space and time, consequently affecting the droplet size distribution. Clustering of larger droplets in the LV region resulted in a higher mean volume radii over there. The most important result from this study
is the different spectral widths ($\sigma$) in the HV and LV regions. In the drier case, a higher value of $\sigma$ occurred in HV region during the first 5 seconds, and after that opposite scenario was observed. This opposite behavior can be connected to droplet evaporation and dilution of number concentration in the HV region. The spectral width always remains higher in the HV area for the moist case (RH=85%), it may be because of higher droplet evaporation influenced by the presence of rotating air pockets, helping to transport the vapor mass out of the HV region.

The intrusion of subsaturated air is the most prominent in the high vorticity regions which reflects in the evolution of the droplet supersaturation. Enhanced evaporation produces a wider droplet and supersaturation spectra. The time series of the droplet number concentration and volume mean radius can be used to get mixing diagrams of high and low vorticity regions. The degree of mixing calculated based on the mixing diagram shows more mixing homogeneity in the HV regions. The Damkohler number which depends upon fluid time scale and phase time scales also indicate a higher degree of homogeneous
mixing in HV regions. Similar observations were found for the mono-disperse case also. We emphasize that our findings are strongly affected by entrainment and mixing of drier air at the cloud edges. The results may differ in adiabatic cloud cores where entrainment and mixing are absent.

*Data availability.* The DNS output data used in this study is archived in HPC Aaditya at IITM pune and can be made available on request.

*Author contributions.* BK and MKY formulated the concept of this work. RR ran simulation and produces results. SB did analysis and
contributed in the manuscript preparation. SAR contributed in the manuscript preparation.

*Competing interests.* The authors have no competing interests.

*Acknowledgements.* The IITM Pune is funded by Ministry of Earth Science (MoES) , Government of India. The simulations were carried out on HPC facility 'Aaditya' (http://aadityahpc.tropmet.res.in/Aaditya/index.html) provided by MoES. This research is partially supported by the NSERC-Hydro-Quebec Industrial Research Chair program to M.K. Yau. The authors would like to thank Mr. Manmeet Singh, CCCR,
IITM for a productive discussion on machine learning algorithms for this work.





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
