# Peer review of "Impact of high and low vorticity turbulence on cloud-environment mixing and cloud microphysics processes"

_Atmospheric Chemistry and Physics, 2021_

## Referee Comment (RC2)

**Review of "Impact of high and low vorticity turbulence on cloud environment mixing and cloud microphysics processes" by Kumar et al. (acp-2021-101)**

The manuscript analyzed the broadening of droplet size distributions in a cloud filament using direct numerical simulation. By analyzing the results for high and low vorticity regions, the authors can show that high vorticity regions cause spectral broadening due to inertia effects, which is a well-known phenomenon, and has been analyzed in a wide range of modeling and observational studies already (Shaw 2003; Grabowski and Wang 2013). However, the novelty of this study lies in the analysis of this process in the context of cloud environment mixing.

Accordingly, I like to support the manuscript's publication in Atmospheric Chemistry and Physics, subject to one major and several minor revisions detailed below. Furthermore, the text contains several spelling and grammar mistakes, which do not impede its comprehensibility, but should be removed in a revised version.

**Major Revisions**

*Criticism of Shaw et al. (1998):* I think the overall criticism of Shaw et al. (1998) in this manuscript is unjustified and misleading. Shaw et al. (1998) laid the foundation for understanding droplet clustering in clouds and its effect on cloud microphysics. Furthermore, Shaw et al. (1998) did not conduct DNS, as falsely claimed in the current manuscript (ll. 54 – 59); they only reviewed previous DNS literature. In fact, Shaw et al. (1998) only used an idealized model based on a Rankine vortex to explain how inertia effects can increase supersaturation, i.e., modeling far from DNS. Moreover, it is very much misleading to claim that the results of the present study are "completely different" from Shaw et al. (1998) (ll. 129 – 133). In fact, the present study actually confirms the original work of Shaw et al. (1998) by showing that clustering is taking place in filaments at the cloud edge and contributes to spectral broadening. Finally, how did the authors determine the "volume fraction of high vorticity" (Tab. 1) that has been used by Shaw et al. (1998)?

**Minor Revisions**

L. 6: No need to defined DNS in the abstract.

Ll. 23 – 24: Although one can see collision and coalescence as two processes, they are often treated as one process, especially in the droplet size range considered in this study. In fact, so do the authors. Therefore, it is awkward to talk about three instead of two processes here.

L. 26: The reference to "Pruppacher 2000" is wrong. The last edition of the book is from 1997, and the authors are Pruppacher and Klett.

Ll. 34 – 35: You should add Grabowski and Wang (2013).

Ll. 39 – 40: "Particle response time" is ambiguous. I prefer "inertial response time" since this is what you mean. A reference on how it is defined would be helpful. Maybe Clift et al. (1978)?

Ll. 43 – 46: The work from Marcia Baker beginning in 1979 with several subsequent publications is missing here (e.g., Baker and Latham 1979).

L. 55: Define DNS.

Ll. 60 – 61: All the recent work by Katarzyna Karpińska and Szymon Malinowski on vortex tubes and their effect on cloud droplet distributions is missing here (e.g., Karpińska et al. 2019).

L. 77: DNS should have been defined already.

Ll. 79 – 81: Why do you mention the output format of your data files? This feels highly irrelevant here.

Ll. 82 – 83: The initial setup is not shown in Fig. 1a.

Ll. 84 – 88, ll. 247 – 255, Tab. 2: I would omit to discuss the monodisperse case in the manuscript. Its discussion feels like an unnecessary addendum to the discussion of the more realistic poly-dispersed

droplet size distribution, without any additional value. Furthermore, it is highly unrealistic to find a monodisperse distribution in cloud edge filaments.

Ll. 84 – 88: Please state that the case with 22 % is unrealistic. In a real cloud, the air in the direct vicinity of a cloud edge filament (< 15 cm) should be moister. However, I think the case still adds to the study since it reveals interesting features of the entrainment-mixing process.

Ll. 84 – 88: Since the poly-dispersed distribution is obtained from measurements, a plot of its shape needs to be provided. Even the cited paper of Kumar et al. (2017) does not contain such a plot.

Ll. 93 – 94: A vortex needs at least 4 grid boxes to be represented on a numerical grid.

Ll. 104 – 107: I do not think it is necessary to state the definition of vorticity here. It is not used in the following, and a simple reference should be sufficient.

Ll. 116 – 117: How do you define a box? Is it a cube? Or a rectangular cuboid? This definition might matter since high vorticity regions are often organized in tubes, which shape might not always fit a "box".

L. 118: "tubular" not "tabular"

L. 116 – 124: How often do you look for high vorticity clusters? Each timestep? Are these high vorticity clusters tracked in time?

L. 134: It is not initially clear that you are discussing the cases with the polydisperse droplet distribution first.

Ll. 138 – 140: What do you mean by these sentences? Please clarify.

L. 142: "The **cloud** volume lies […]"

Fig. 3: If the slab and the edge occupy the same volume of the model domain, how is it possible to have different kinetic energies at the beginning of the simulation? I would assume that the initial kinetic energy is uniformly distributed since evaporation processes only start after the beginning of the simulation. The same applies to the mean vorticity.

L. 159: Clarify the qv is the water vapor mixing ratio.

Fig. 4: At what size is a particle considered a droplet? Or, in other words, how do you determine Nd?

L. 194: How do you calculate the spectral width? There are several ways to do it, and they can result in significant differences.

Ll. 194 – 197, Fig. 5: An additional plot showing the dispersion of the droplet size distribution ($\sigma_r/r_{mean}$) is necessary to make the statement about different droplet size widths more robust! Since the mean radius is also changing significantly between the high and low vorticity regions, changes in the spectral width are inevitable, even in the absence of a process that contributes to broadening.

Ll. 206 – 208: A reference to Tölle and Krueger (2014) or Luo et al. (2020) might be appropriate here.

L. 219, and several other places: It is odd to use the word supersaturation when you write about subsaturations. I suggest using "saturation ratio" instead.

Fig. 6: State the time at which these spectra are calculated in the plots.

Ll. 224 – 229: This is a great paragraph. It shows clearly that high vorticity regions can be identified as zones of entrainment. I would state this more explicitly.

Fig. 7, lower panel: The label on the ordinate of the plot should read $\sigma_S$ and not $\sigma_r$. Is $\sigma_S$ shown in percent?

Ll. 234 – 235: You are writing about Fig. 8, not Fig. 7.

L. 235: How do you define the degree of homogeneous mixing? Do you use the approach by Morrison and Grabowski (2008)?

L. 235 and several other places: Damk**ö**hler not Damkohler.

Ll. 237 – 240: It might be more appropriate to use the droplet evaporation timescale here since you are interested in changes in the droplet size distribution and not the thermodynamics.

Fig. 8: The entire discussion is based on panel d, although panels a to c provide valuable information. What does "line at nh" (panels a and b) mean?

L. 280: Define urms.

Ll. 282 – 283: Where do we see this in the current study? I think this statement is true, but the presented results do not confirm this directly.

L. 285: Change $\sigma$ to $\sigma_r$ in accordance with the notation of the manuscript.

L. 287 – 289: This is interesting. However, it is probably also the increased transport of less-moist air into these regions, i.e., entrainment of environmental air, which you are observing.

**Technical Corrections**

There are many spelling and grammatical errors which need to be addressed.

**References**

Baker, M. B., & Latham, J. (1979). The evolution of droplet spectra and the rate of production of embryonic raindrops in small cumulus clouds. *Journal of the Atmospheric Sciences*, *36*(8), 1612-1615.

Clift, R., Grace, J. R., & Weber, M. E. (2005). Bubbles, drops, and particles.

Grabowski, W. W., & Wang, L. P. (2013). Growth of cloud droplets in a turbulent environment. *Annual review of fluid mechanics*, *45*, 293-324.

Karpińska, K., Bodenschatz, J. F., Malinowski, S. P., Nowak, J. L., Risius, S., Schmeissner, T., ... & Bodenschatz, E. (2019). Turbulence-induced cloud voids: observation and interpretation. *Atmospheric Chemistry and Physics*, *19*(7), 4991-5003.

Kumar, B., Bera, S., Prabha, T. V., & Grabowski, W. W. (2017). Cloud-edge mixing: Direct numerical simulation and observations in I ndian M onsoon clouds. *Journal of Advances in Modeling Earth Systems*, *9*(1), 332-353.

Luo, S., Lu, C., Liu, Y., Bian, J., Gao, W., Li, J., ... & Guo, X. (2020). Parameterizations of Entrainment-Mixing Mechanisms and Their Effects on Cloud Droplet Spectral Width Based on Numerical Simulations. *Journal of Geophysical Research: Atmospheres*, *125*(22), e2020JD032972.

Morrison, H., & Grabowski, W. W. (2008). Modeling supersaturation and subgrid-scale mixing with two-moment bulk warm microphysics. *Journal of the Atmospheric Sciences*, *65*(3), 792-812.

Shaw, R. A., Reade, W. C., Collins, L. R., & Verlinde, J. (1998). Preferential concentration of cloud droplets by turbulence: Effects on the early evolution of cumulus cloud droplet spectra. *Journal of the atmospheric sciences*, *55*(11), 1965-1976.

Shaw, R. A. (2003). Particle-turbulence interactions in atmospheric clouds. *Annual Review of Fluid Mechanics*, *35*(1), 183-227.

Tölle, M. H., & Krueger, S. K. (2014). Effects of entrainment and mixing on droplet size distributions in warm cumulus clouds. *Journal of Advances in Modeling Earth Systems*, *6*(2), 281-299.

---

## Author Comment (AC1)

**Response to reviewer #1**

We thank the reviewer for taking the time to read our manuscript and for providing the comments. The responses are provided below in blue color.

**Overview:**
In this paper, analysis of the thermodynamic and microphysical characteristics of droplets and flow in high and low vorticity regions. The study performed direct numerical simulation of turbulent flow with droplet evaporation/condensation in a sub-meter cubed sized domain. The topic is interesting and the manuscript requires little improvement, especially the correction of grammatical mistakes. The introduction provides a good and concise (theoretical) background to the study.

**Response:** It is nice to know that reviewer found our work interesting.

The scientific merit of the study deserves publication. Yet, I recommend minor revision of the manuscript before its acceptance. This recommendation is based on the comments and remarks listed below:

**Response:** Thank you very much to the reviewer for recommending acceptance of the manuscript with minor revision for publication in the journal ACP.

1. This work is exceptional for including the entrainment-mixing and resolving the Kolmogorov time scales but I am wondering why the authors chose k = 3500 as the optimal k value. I will suggest that the authors try larger values of k in figure 1c. Why is the maximum number of iteration chosen as 200?

**Answer:**

Why 'k=3500'

Vortices have tubular or sheet like structures. So, a 3D box enclosing a vortex may also include many low vorticity points. If we make the boxes smaller (which is done by increasing "n_clusters"), fewer number of low vorticity points are included in the boxes. At k=3500, the average vorticity in the boxes obtained reach the selected threshold vorticity (60 s-1), as shown in figure 1 below. This figure is already included in the manuscript (Figure 1c).

With increasing value of 'k', the size of the clusters decreases. Some clusters may become so small that they will include two or three (say) high vorticity points only, all in the same plane. Therefore, the 'k=3500' value was found to be optimal. If we take high value of 'k' then we may get many zero volume boxes. That's why increasing the value of 'k' indefinitely is not advisable.

Why 200 iteration:

The optimal numbers of iterations were chosen as 200 to keep the computational cost manageable.

[Figure]

**Figure 1:** Average vorticity of 3D boxes for different value of "n_clusters".

2. In figure 3, I guess the mean KE and vorticity is averaged over the slab or edge volume. It should written in the caption.

**Answer:**
The average was taken over the cloudy slab and both edges. We have included this information in the figure caption in the revised manuscript.

3. In line 159-160, the authors wrote that they investigated the evolution of the mixing ratio but there is no figure showing the evolution of the mixing ratio and the u_{rms}.

**Answer**: The figures are shown below.

[Figure]

Figure 2: The evolution of mixing ratio and U$_{rms}$.

We have not provided the figure in the manuscript because we wanted to report the analysis results only. We have modified the text in the revised manuscript.

4. In the introduction, the authors did not explicit write the scientific questions for this study. It is written in the conclusion. This can be confusing for the reader

**Answer:**
Done. We have added text in the introduction section (at the end) addressing the scientific questions.

5. What is the time step for the simulation? Can you present the energy spectrum for the flow field?

**Answer:** The time step was 0.0005 seconds? The energy spectrum is provided in figure 3.

[Figure]

Figure 3: The energy spectrum for the fluid flow.

6. In line 82-83, the authors wrote that "an initial setup of computational domain is presented by the Figure 1(a)". Figure 1(a) does not contain the initial setup. Are you referring to figure 1(d)?

   **Answer:** This is a typo. The correct one is figure 2. We have updated the text at this location by refereeing to section 3 for the initial set up.

7. The authors wrote that the mono-dispersed droplet size distribution cases are idealized cases. These idealized cases should have been discussed first before the poly-dispersed cases. Why? The authors gave a short summary of these idealized cases in section 4 and table 2 with no figure to substantiate the conclusions in table 2.

   Answer: We agree with the reviewer and thank he/she for pointing this out. Since it is not adding any value in this work, we have removed the discussion of the mono-dispersed distribution case in the updated manuscript.

**Minor corrections**

1. In line 69, change "We compared …" to "We compare…"

**Response:** Done

2. In line 72, change "we aims to look …" to "we aim to look…".  Also, change "section provides details of methods employed …" to "section provides the details of all methods and data used"

**Response:**
 Done. We have added a few lines to address the scientific questions as suggested by the reviewer.

3. In line 83, change "is presented by …" to "is presented in …"

**Response:** Done

4. This sentence "The next step is to find …" in line 92-93 should be rewritten. I will suggest you break this sentence into two.

**Response:** Done. The sentence is broken in two parts.

5. I will suggest the authors get a professional to correct all grammatical mistakes in the manuscript.

**Response:** Used professional software for checking grammar. The paper is also edited to improve the readability.

---

## Author Comment (AC3)

**Response to reviewer #2**

First of all we would like to thank to the reviewer for providing the comments on our manuscript. Our responses are provided in this document. We have put the response in the blue color text while the reviewer's comments are in black color.

**Review of "Impact of high and low vorticity turbulence on cloud environment mixing and cloud microphysics processes" by Kumar et al. (acp-2021-101)**

The manuscript analyzed the broadening of droplet size distributions in a cloud filament using direct numerical simulation. By analyzing the results for high and low vorticity regions, the authors can show that high vorticity regions cause spectral broadening due to inertia effects, which is a well known phenomenon, and has been analyzed in a wide range of modeling and observational studies already (Shaw 2003; Grabowski and Wang 2013). However, the novelty of this study lies in the analysis of this process in the context of cloud environment mixing.

Accordingly, I like to support the manuscript's publication in Atmospheric Chemistry and Physics, subject to one major and several minor revisions detailed below. Furthermore, the text contains several spelling and grammar mistakes, which do not impede its comprehensibility, but should be removed in a revised version.

**Response:** Thanks a lot for supporting our manuscript for publication.

**Major Revisions**
*Criticism of Shaw et al. (1998):* I think the overall criticism of Shaw et al. (1998) in this manuscript is unjustified and misleading. Shaw et al. (1998) laid the foundation for understanding droplet clustering in clouds and its effect on cloud microphysics. Furthermore, Shaw et al. (1998) did not conduct DNS, as falsely claimed in the current manuscript (ll. 54 – 59); they only reviewed previous DNS literature. In fact, Shaw et al. (1998) only used an idealized model based on a Rankine vortex to explain how inertia effects can increase supersaturation, i.e., modeling far from DNS. Moreover, it is very much misleading to claim that the results of the present study are "completely different" from Shaw et al. (1998) (ll. 129 – 133). In fact, the present study actually confirms the original work of Shaw et al. (1998) by showing that clustering is taking place in filaments at the cloud edge and contributes to spectral broadening.

**Response**: We thank the reviewer for pointing out this mistake.

The motivation of this study is not to criticise the results of Shaw et al. (1998) rather to present a comparison with previous works (i.e., Shaw et al.). We agree with the reviewer that it was wrongly claimed that the Shaw et al.'s work is a DNS study. In the revised manuscript, we have corrected this mistake and modified the sentences which appeared a criticism of Shaw et al. (1998).

AS suggested by the reviewer, Shaw et al. (1998) laid the foundation for droplet clustering in clouds and the effect on DSD. However, there are many differences between the simulation of Shaw et al. and the present DNS simulation. In Figure 1b, Shaw et al. (1998) showed a significant amount of droplet clustering (which is based on the DNS study of Squires and Eaton, 1991) of about 50 % using Rankine vortex in their parcel model. In contrast, our DNS generates about 0.2% of volume fraction of high vorticity and is in agreement with the comments in Vaillancourt and Yau (2000). In Shaw et al., the assumption of very high fraction vortex structure might lead to significant spectral broadening which was also pointed out by Grabowski and Vaillancourt (1999).

The preferential concentration of droplets was several folds higher than the mean droplet concentration as assumed by Shaw et al. (originally reported by DNS of Squires and Eaton, 1991). However, here we can see droplet concentration in high vorticity region not higher than 1.5 fold of the low vorticity region. The spectral broadening in Shaw et al. was contributed by supersaturation fluctuation in low and high vorticity regions and the combined effect of low and high vorticity. However, in the present case high vorticity region produced wider spectral broadening compared to low vorticity region.

Another important aspect is that in Shaw et al. (1998), the droplet growth model is based on supersaturation difference between HV and LV regions resulted by droplet concentration difference. They also did not considered the effect of droplet size difference as larger droplets are more prone to be flown out of HV region and accumulate in LV region which is accounted for in present study.

 We have clarified this issue in the revised manuscript.

How did the authors determine the "volume fraction of high vorticity" (Tab. 1) that has been used by Shaw et al. (1998)?
Response: Shaw et al. constructed two zones of vortices (high and low vorticity) in their parcel model a with Rankin vortex and considered the same volume fraction (i.e. 50%) for the two vorticity zones as stated on P-1971, right side second paragraph.

**Minor Revisions**
L. 6: No need to defined DNS in the abstract.
Answer: Done. I have removed it.
Ll. 23 – 24: Although one can see collision and coalescence as two processes, they are often treated as one process, especially in the droplet size range considered in this study. In fact, so do the authors. Therefore, it is awkward to talk about three instead of two processes here.
Answer: Done: We made it two processes instead of three.

L. 26: The reference to "Pruppacher 2000" is wrong. The last edition of the book is from 1997, and the authors are Pruppacher and Klett.

Answer: Corrected.

Ll. 34 – 35: You should add Grabowski and Wang (2013).

Answer: Added

Ll. 39 – 40: "Particle response time" is ambiguous. I prefer "inertial response time" since this is what you mean. A reference on how it is defined would be helpful. Maybe Clift et al. (1978)?

Answer: Done.

Ll. 43 – 46: The work from Marcia Baker beginning in 1979 with several subsequent publications is missing here (e.g., Baker and Latham 1979).

Answer: Added (Baker and Latham 1979)

L. 55: Define DNS.

Answer: Defined it later on line 69 in the updated manuscript.

Ll. 60 – 61: All the recent work by Katarzyna Karpińska and Szymon Malinowski on vortex tubes and their effect on cloud droplet distributions is missing here (e.g., Karpińska et al. 2019).
Answer: Added

L. 77: DNS should have been defined already.

Answer: Removed from this line and defined before.

Ll. 79 – 81: Why do you mention the output format of your data files? This feels highly irrelevant here.

Answer: It is data section. We are talking about some methods for data analysis. As such, it is important to mention the format of the data used. For instance, reading the NetCDF format data is easier in visualization tools as it can be directly read. On the other hand, the SION format requires data pre-processing to make it readable by that software.

Ll. 82 – 83: The initial setup is not shown in Fig. 1a.

Answer: This was a typo. It has been corrected. The initial set up is shown in Fig 2 in section 3. This information is included in the revised manuscript.

Ll. 84 – 88, ll. 247 – 255, Tab. 2: I would omit to discuss the monodisperse case in the manuscript. Its discussion feels like an unnecessary addendum to the discussion of the more realistic poly-dispersed droplet size distribution, without any additional value. Furthermore, it is highly unrealistic to find a monodisperse distribution in cloud edge filaments.

Answer: We agree with the reviewer. We omitted the discussion on mono-disperse case.

Ll. 84 – 88: Please state that the case with 22 % is unrealistic. In a real cloud, the air in the direct vicinity of a cloud edge filament (< 15 cm) should be moister. However, I think the case still adds to the study since it reveals interesting features of the entrainment-mixing process.
Answer: The case RH22% humidity was considered to see the effect of entrainment and mixing process in high and low vorticity regions.

Ll. 84 – 88: Since the poly-dispersed distribution is obtained from measurements, a plot of its shape needs to be provided. Even the cited paper of Kumar et al. (2017) does not contain such a plot.

Answer: We have given the range of droplet sizes for the distribution. We have added the information of the mean radii in this distribution. We think including one more figure is not needed. Such a plot is available in Fig 1. of Kumar et al. (2017).

Ll. 93 – 94: A vortex needs at least 4 grid boxes to be represented on a numerical grid.

Answer: Yes, agree.  We mean to say here that, we calculated the vorticity magnitude from the raw Eulerian data outputted from the simulation. This data has velocity values at each grid points. We first generated the data having vorticity magnitudes at those grid points. Thereafter, we sought the boxes in the domain which can contain a vortex.  We have explained this in subsequent lines.

Ll. 104 – 107: I do not think it is necessary to state the definition of vorticity here. It is not used in the following, and a simple reference should be sufficient.

Answer: We agree. However, this information was asked in the initial round of reviews from the editorial board to make smooth flow for the readers.

Ll. 116 – 117: How do you define a box? Is it a cube? Or a rectangular cuboid? This definition might matter since high vorticity regions are often organized in tubes, which shape might not always fit a "box".
Answer: We have considered a cubical boxes or a cuboid to cover a particular high vorticity region. Please see the  figure 1(d). We have illustrated three such boxes (red color) in that panel.

L. 118: "tubular" not "tabular"
Answer: Thanks. It was a typo. We have corrected it.

L. 116 – 124: How often do you look for high vorticity clusters? Each timestep? Are these high vorticity clusters tracked in time?

Answer: Yes, we look for the clusters at every time step. However, the clusters are not tacked on time rather we just find the high vorticity regions based on a threshold value.

L. 134: It is not initially clear that you are discussing the cases with the polydisperse droplet distribution first.

Answer: We have made it clear in Data and Methods section in the modified manuscript L. 91).

Ll. 138 – 140: What do you mean by these sentences? Please clarify.

Answer: We mean to say that we divided the whole computational domain in to three distinct parts to illustrate the cloudy slab and the cloud interfaces. These parts are illustrated in figure 2.

L. 142: "The **cloud** volume lies […]"
Fig. 3: If the slab and the edge occupy the same volume of the model domain, how is it possible to have different kinetic energies at the beginning of the simulation? I would assume that the initial kinetic energy is uniformly distributed since evaporation processes only start after the beginning of the simulation. The same applies to the mean vorticity.

Answer:
It is not only the droplet evaporation that generates instability but it is the interface instability due to the density gradient at the cloud edge that generates instability when switching on the simulation and thereby producing more KE and vorticity. The cloud slab is kept in the middle of domain (see the figure 2) and hence, the entrainment happens from the two sides of the slab.

L. 159: Clarify the qv is the water vapor mixing ratio.

Answer: Yes, it is. We have modified the text in the manuscript.

Fig. 4: At what size is a particle considered a droplet? Or, in other words, how do you determine Nd?
Answer: The particle above 1 µm  size were considered as droplets. Number density was calculated based on total number of droplets in the whole domain and the total volume of the domain.

Ll. 194 – 197, Fig. 5: An additional plot showing the dispersion of the droplet size distribution ($\sigma!/r_{mean}$) is necessary to make the statement about different droplet size widths more robust! Since the mean radius is also changing significantly between the high and low vorticity regions, changes in the spectral width are inevitable, even in the absence of a process that contributes to broadening.

Answer:  The relative dispersion graph has been included in figure 5.

Ll. 206 – 208: A reference to Tölle and Krueger (2014) or Luo et al. (2020) might be appropriate here.

Answer: Done.  The latest reference has been added.

L. 219, and several other places: It is odd to use the word supersaturation when you write about subsaturations. I suggest using "saturation ratio" instead.

Answer: Done.   The word "subsaturation" has been changed at most of the places where it was appropriate.

Fig. 6: State the time at which these spectra are calculated in the plots.

Answer: It is stated in the caption of the figure. The statement is "Upper panel depicts the plots for 3 seconds and the bottom one is for 17.8 seconds."

Ll. 224 – 229: This is a great paragraph. It shows clearly that high vorticity regions can be identified as zones of entrainment. I would state this more explicitly.

Answer: Thanks a lot.  The said statement has been added at the end of the paragraph.

Fig. 7, lower panel: The label on the ordinate of the plot should read $\sigma_s$ and not $\sigma_r$. Is $\sigma_s$ shown in percent?

Answer: Yes. Thanks for this corrections. We have updated the ordinate of figure 7.  The $\sigma_s$ is in percent.

Ll. 234 – 235: You are writing about Fig. 8, not Fig. 7.

Answer: Yes. It was a typo, now corrected in the revised manuscript.

L. 235: How do you define the degree of homogeneous mixing? Do you use the approach by Morrison and Grabowski (2008)?

Answer:

We used method provide by Lu et al. (2012) where they defined three measures of homogeneous-mixing degree. . In this study, we adopt the first method which is based on a 'β' parameter where,

$$\beta = \tan^{-1}\left\{\frac{\frac{r^3}{r_a^3}-1}{\frac{n}{n_a}-\frac{n_h}{n_a}}\right\}$$

β is normalized by $\frac{\pi}{2}$ to get the measure of homogeneous mixing degree or ψ.

 Where

r = volume mean radius
$r_a$ = adiabatic volume mean radius
n = number concentration

$n_h$ = homogeneous number concentration
$n_a$ = adiabatic number concentration

This information has been added in the revised manuscript.

L. 235 and several other places: Damk**ö**hler not Damkohler.

Answer: Corrected.

Ll. 237 – 240: It might be more appropriate to use the droplet evaporation timescale here since you are interested in changes in the droplet size distribution and not the thermodynamics.

Answer: We calculated the Damköhler number based on evaporation timescale. The evaporation timescale has been calculated taking reference from Andrejczuk et al. (2009) and Bera et al. (2016). See the figure below.

[Figure]

Fig1 : The Damköhler number based on  evaporation timescale

However, we kept the original figure of Damköhler number based on phase relaxation timescale as we believe that evaporation time only dictates during early evolution of the simulation when there exist a cloud free environment. But after a few seconds there is no cloud free environment exists in the domain as droplets spreads over entire domain. So, the phase relaxation time is more important during later part of evolution. However, the present results are not going to change even if we consider the τevaporation as we can see in figure 1 above which indicates the mixing during early evolution even based on τevaporation.

**Fig. 8**: The entire discussion is based on panel d, although panels a to c provide valuable information. What does "line at nh" (panels a and b) mean?

Answer: We have provided the discussion for other panels and clarify the meaning of the line at $n_h$ in the revised manuscript. The details are provided below.

In the mixing diagram, the volume mean radius cube ($r^3$) is plotted against the number concentration ($N_d$). Sometimes, a normalized version, in which the radius and number concentration are normalized by the respective adiabatic values (Gerber et al. (2008), Kumar et al. (2014). In the same way, we calculated the mixing diagrams which are depicted in Figure 8(a & b). Here, we have considered the initial values (t=0s) as adiabatic in the mixing diagram. The mixing diagrams, for both RH cases, show a clear picture of mixing types. In the low vorticity case, the mixing is inactive till 1.2 seconds while for HV it goes up to 1.4 seconds. During this time, entrained air just dilutes the number concentration ($n_h$) without acting on droplet evaporation (because the cloudy slab expands during this time and droplets scatter in to the entire domain). Afterward, physical mixing occurs when the mixing diagram takes a turn toward the homogenous mixing regime where both the number density and mean volume radius decrease rapidly. One can summarize that in HV regions the mixing remains more homogeneous compare to LV regions as can be seen by more rapid droplet evaporation (decrease in $r$) than the decrease in number density.

The degree of mixing based on the study of Lu et al. (2012) is presented in Figure 8(c). It is calculated using equation 3 {Lu_2012}. The values of the degree lies in range of [0, 1]. The value 1 represents homogeneous mixing and extreme inhomogeneous mixing has value zero. We observed a higher mixing degree in the HV regions in both RH cases. The moist case shows relatively inhomogeneous mixing initially for both LV and HV regions and gradually it moves towards homogeneous mixing although the HV region always remains at the higher side of homogeneous mixing.

L. 280: Define urms.

Answer: We have corrected it.

Ll. 282 – 283: Where do we see this in the current study? I think this statement is true, but the presented results do not confirm this directly.

Answer: It is clear from the number density analysis in section 3.4.1 and figure 4. The number density and mean radius was found higher in the LV region.

L. 285: Change $\sigma$ to $\sigma_r$ in accordance with the notation of the manuscript.

Answer: Done.

L. 287 – 289: This is interesting. However, it is probably also the increased transport of less-moist air into these regions, i.e., entrainment of environmental air, which you are observing.

Answer: Yes, we agree with the reviewer.
We mean to say the transportation of droplets out of HV regions. However, as stated, it is also true that entrainment of environmental air into these regions enhance the droplet evaporation and consequently, supporting higher spectral width.
We have added a sentence in the revised manuscript as suggest by you.

**Technical Corrections**

There are many spelling and grammatical errors which need to be addressed.

Answer: Used professional software to check these errors. The manuscript is also edited for better readability.

**References**

Kumar, B., Schumacher, J. and Shaw, R. A. : Lagrangian Mixing Dynamics at the Cloudy-Clear Air Interface, J. Atmos. Sci., 71(7), 2564–2580. doi: 10.1175/JAS-D-13-0294.1,2014.

Hermann E. GERBER, Glendon M. FRICK, Jorgen B. JENSEN, James G. HUDSON, Entrainment, Mixing, and Microphysics in Trade-Wind Cumulus, Journal of the Meteorological Society of Japan. Ser. II, 2008, Volume 86A, Pages 87-106,

Baker, M. B., & Latham, J. (1979). The evolution of droplet spectra and the rate of production of embryonic raindrops in small cumulus clouds. *Journal of the Atmospheric Sciences*, *36*(8), 1612-1615.

Clift, R., Grace, J. R., & Weber, M. E. (2005). Bubbles, drops, and particles.

Grabowski, W. W., & Wang, L. P. (2013). Growth of cloud droplets in a turbulent environment. *Annual review of fluid mechanics*, *45*, 293-324.

Karpińska, K., Bodenschatz, J. F., Malinowski, S. P., Nowak, J. L., Risius, S., Schmeissner, T., ... & Bodenschatz, E. (2019). Turbulence-induced cloud voids: observation and interpretation. *Atmospheric Chemistry and Physics*, *19*(7), 4991-5003.

Kumar, B., Bera, S., Prabha, T. V., & Grabowski, W. W. (2017). Cloud-edge mixing: Direct numerical simulation and observations in Indian Monsoon clouds. *Journal of Advances in Modeling Earth Systems*, *9*(1), 332-353.

Luo, S., Lu, C., Liu, Y., Bian, J., Gao, W., Li, J., ... & Guo, X. (2020). Parameterizations of Entrainment-Mixing Mechanisms and Their Effects on Cloud Droplet Spectral Width Based on Numerical Simulations. *Journal of Geophysical Research: Atmospheres*, *125*(22), e2020JD032972.

Shaw, R. A., Reade, W. C., Collins, L. R., & Verlinde, J. (1998). Preferential concentration of cloud droplets by turbulence: Effects on the early evolution of cumulus cloud droplet spectra. *Journal of the atmospheric sciences*, *55*(11), 1965-1976.

Shaw, R. A. (2003). Particle-turbulence interactions in atmospheric clouds. *Annual Review of Fluid Mechanics*, *35*(1), 183-227.

---

## Referee Report (RR1)

**Review of "Impact of high and low vorticity turbulence on cloud environment mixing and cloud microphysics processes" by Kumar et al. (acp-2021-101)**

The authors have addressed most of my comments with care. While I have some very minor comments below, the manuscript is generally ready for publication. I do not need to see it again.

Line numbers refer to the tracked-changes version of the manuscript.

**Minor Comments**

Ll. 91 – 94, 294, 331: Since the mono-disperse case is not discussed in the revised version, I suggest removing all remaining references. (Alternatively, one should state clearly in the conclusions section that the mono-disperse case is not discussed in the paper.)

Ll. 143 – 145: I fully agree that the volume fraction of 50 % is unrealistic. However, assessing the volume fraction of high vorticity regions has never been an objective of Squires and Eaton (1991) and Shaw et al. (1998). Therefore, I suggest writing "assumed high volume fraction" instead of "finding of high volume fraction".

L. 171: Writing about a (singular) instability feels odd. I suggest using the plural ("instabilities").

Ll. 218 – 220: By showing the relative dispersion in Fig. 6, one could strengthen the argument here: Broadening is always stronger in the high vorticity regions.

Ll. 263 – 269: Technically speaking, the number density should be constant in the homogeneous mixing regime. Accordingly, the statement "a turn toward the homogeneous mixing regime where both the number density and the mean volume radius decrease rapidly" is slightly misleading.

**Technical Corrections**

L. 6: Replace "DNS" with "direct numerical simulation".

L. 137: "the required cuboid" or "a cuboid"

Caption of Fig. 3: "KE", not "K.E."

L. 264: "till" is informal.

Ll. 263 – 264: Please rewrite this sentence. (Maybe: "In the LV case, mixing is inactive for the first 1.2 s, and for 1.4 s in the HV case.)

---

## Author Response (AR2)

**Review of "Impact of high and low vorticity turbulence on cloud environment mixing and cloud microphysics processes" by Kumar et al. (acp-2021-101)**

The authors have addressed most of my comments with care. While I have some very minor comments below, the manuscript is generally ready for publication. I do not need to see it again.

Line numbers refer to the tracked-changes version of the manuscript.

**Dear Reviewer,**
**Thank you very much for giving time to review our manuscript and recommending for publication with minor revisions. Our responses the provided below in blue color.**

**Minor Comments**
Ll. 91 – 94, 294, 331: Since the mono-disperse case is not discussed in the revised version, I suggest removing all remaining references. (Alternatively, one should state clearly in the conclusions section that the mono-disperse case is not discussed in the paper.)

Response: We have modified the sentence in line 91-94, see the red color in modified manuscript file with track change. Also removed the discussion on mono-disperse case from conclusion section.

Ll. 143 – 145: I fully agree that the volume fraction of 50 % is unrealistic. However, assessing the volume fraction of high vorticity regions has never been an objective of Squires and Eaton (1991) and Shaw et al. (1998). Therefore, I suggest writing "assumed high volume fraction" instead of "finding of high volume fraction".

Response: Done. See the red color text.

L. 171: Writing about a (singular) instability feels odd. I suggest using the plural ("instabilities").

Response: Changed. See line 173 in the tracked changed version.

Ll. 218 – 220: By showing the relative dispersion in Fig. 6, one could strengthen the argument here: Broadening is always stronger in the high vorticity regions.

Response: Added this information (line 222) in the modified manuscript.

Ll. 263 – 269: Technically speaking, the number density should be constant in the homogeneous mixing regime. Accordingly, the statement "a turn toward the homogeneous mixing regime where both the number density and the mean volume radius decrease rapidly" is slightly misleading.

Response: The statement has been revised.

**Technical Corrections**

L. 6: Replace "DNS" with "direct numerical simulation".

Response: Done

L. 137: "the required cuboid" or "a cuboid"

Response: Done

Caption of Fig. 3: "KE", not "K.E."

Response: Done

L. 264: "till" is informal.

Response: Changed with 'up to'.

Ll. 263 – 264: Please rewrite this sentence. (Maybe: "In the LV case, mixing is inactive for the first 1.2 s, and for 1.4 s in the HV case.)

Response: Done. The sentence has been modified.